

# The building, shaping, and filling of an Upper Slope Terrace: the Rio Grande Terrace, SW Atlantic

Michel Michaelovitch de Mahiques[1,2], Javier Alcántara-Carrió[1], Francisco José Lobo[3], Uri Schattner[4], Rosangela Felício dos Santos[1], Samara Cazzoli y Goya[1], Raissa Basti Ramos[1] José Gustavo Natorf de Abreu[5], Luiz Antonio Pereira de Souza[6], Rubens Cesar Lopes Figueira[1], Marcia Caruso Bícego[1]

[1] Oceanographic Institute, University of São Paulo, São Paulo, 05508-120, Brazil
[2] Institute of Energy and Environment, University of São Paulo, São Paulo, 05508-010, Brazil
[3] Instituto Andaluz de Ciencias de la Tierra, Armilla, 18100, Spain
[4] Leon H Charney School of Marine Sciences, University of Haifa, Haifa, 3498838, Israel
[5] Escola do Mar, Ciência e Tecnologia, UNIVALI, Itajaí, 88302-901, Brazil
[6] Instituto de Pesquisas Tecnológicas do Estado de São Paulo, São Paulo, 05508-901, Brazil

*Correspondence to*: Michel M. de Mahiques (mahiques@usp.br)

**Abstract.** In this work, we analyze the origin, evolution, and recent sedimentation of the Rio Grande Terrace (RGT), the most prominent upper slope terrace of the SW Atlantic. We gathered data from deep to shallow seismic, echo-sounding and surface sediments in order to recognize the role played by the geological inheritance, the sediment origin, and the present hydrodynamics, this last one defined by the action of the high speeds of the Brazil Current.

The RGT is here interpreted as a unique geomorphological feature, which was developed over a pre-Aptian basement high. It presents a complex modern morphology, given by the erosional action of the Brazil Current and its present surface sediments show a high complexity in terms of origin and grain size.

## 1 Introduction

Upper continental slope sand deposits occur in different environments, such as carbonate reefs (Harris and Davies, 1989; Lidz, 2006) or previously glaciated areas (O'Reilly et al., 2007), formed or not in phases of sea-level stabilization (Øvrebø et al., 2006). They can be formed in active (Klaeschen et al., 1994; Dobson et al., 1998; Davy and Collot, 2000) or passive (Kenyon, 1986; Weaver et al., 2000) tectonic margins.

On passive, siliciclastic margins, the development of upper slope sand deposits has been associated to the interaction of superficial boundary currents with the ocean bottom (Viana and Faugères, 1998; Viana et al., 1998; Viana et al., 2002). Rodriguez and Anderson (2004) identified an extensive sand deposit on the Antarctica outer shelf and upper slope, formed in the last ca. 9,000 years, and associated it to the action of the Circumpolar Deep Water.



In the case of the SW Atlantic slope, the maintenance of these features was attributed to the "floor-polisher" effect of the Brazil Current (de Mahiques et al., 2002; Viana et al., 2002), forming upper slope terraces. Hernández-Molina et al. (2008) refer these terraces as one of the six different large-scale erosional features associated with contourites. According to Kowsmann and Carvalho (2001), these terraces have been formed during the last lowstand and permitted the removal of

approximately 40 meters of sediment in the Campos Basin.

Deeper terraces also occur in the world margins, being associated either with tectonic controls (Hernández-Molina et al., 2006) or with the interaction between moving water masses and currents (Preu et al., 2013; Hernández-Molina et al., 2015).

On the SW Atlantic upper margin, a series of erosion terraces occur, in which the most prominent is the Rio Grande Terrace (RGT) (Zembruscki, 1979; Bassetto et al., 2000). Until the present days, the only papers that dealt with this important

geomorphological feature are related almost exclusively to the occurrence of phosphorite deposits on its northern part (Cooke et al., 2007; Pinho et al., 2011; Abreu et al., 2014; Pinho et al., 2015).

This study aims to provide the first characterization of the RGT, regarding its morphostructure and recent sedimentation, as an attempt to recognize the role of geological heritage and modern hydrodynamics in its present configuration.

**2 Study Area**

The RGT (Fig. 1) is a 5,500 km2 erosional terrace located between the latitudes 29º00'S and 31º00'S and longitudes 47º30'W and 48º30'W (Zembruscki, 1979; Bassetto et al., 2000). Its first morphological characterization was done by Zembruscki (1979), based on echosounder records from the 1960s and 1970s. The author defined the RGT as a horizontal feature located over the Torres Arch (Rabinowitz and LaBrecque, 1979; Meisling et al., 2001), positioned between the shelf break and the

lower slope. The author also described the morphology as marked by a rough surface, cut by valleys.

The limited information about its sedimentary structure and infill is based on the regional works by Kowsmann et al. (1977), Bassetto et al. (2000), and Stica et al. (2014).

Kowsmann et al. (1977) analyzed seismic refraction lines from southern Brazil, including some near the RGT (Fig. 1 and 2). The authors identified a high-velocity (> 6.0 km.s-1) shallow feature on the area, interpreting it as a dike, associated with the

Torres Arch (Gladczenko et al., 1997). A whole sequence from Lower Cretaceous to Pliocene sediments was interpreted on the base of acoustic and well log data (Fig. 2A and B).

Stica et al. (2014), using deep-seismic and magnetic anomalies values, interpreted the occurrence of a volcanic dome, positioned just below the Terrace, deforming rift sediments. This dome should correspond to the same high-velocity feature described by Kowsmann et al. (1977).

Apart of the papers by Pinho et al. (2011), Pinho et al. (2015) and Figueiredo and Tessler (2004), there is no information about the sedimentary cover in the area. The first two papers used echosounder backscatter values to describe the prevalence of high Bottom Surface Backscattering Strength (BSBS) values in the area, associating them with the abundance of



phosphorite and carbonate-rich bottoms. Information about grain size and calcium carbonate content, available in Figueiredo and Tessler (2004) is based on very few samples, reporting the occurrence of carbonate-rich fine to medium sands.

Concerning information on circulation, there is no direct information about current speed and direction in the area. Nevertheless, since the RGT is located to the south of the Santos Bifurcation, it is admissible a general southward flow of the current systems, from the outer shelf to the lower slope (Boebel et al., 1999; Schmid et al., 2000; Valla et al., 2018).

There is no adjacent river with significant input of sediments and a single sediment sample (578 m water depth) located farther north, and analyzed by Razik et al. (2015), indicates a mixed influence of the Plata mudbelt (Uruguayan shelf), together with the Pampean (Argentina) shelf, as potential sediment sources to the area.

## 3 Methods

The data used in this work is based on previously obtained acoustic lines and samples, as well as on information collected during an oceanographic survey on board R.V. Alpha Crucis, belonging to the University of São Paulo (Brazil), held in November-December 2017. All of the information regarding data is summarized in Fig.1.

Eight lines of high-resolution single-beam bathymetry were provided by the Brazilian Navy, under the auspices of the Program of Recognition of the Jurisdictional Continental Shelf (Mohriak and Torres, 2017). Each geographical coordinate was transformed to UTM values (WGS84), and depth versus distance graphs were drawn. Changes in declivity were calculated using the mobile mean of ten sequential echo-sounding data.

Eight 2D time-migrated multi-channel seismic reflection profiles were released for academic use by the National Agency for Oil, Natural Gas, and Biofuels (ANP-Brazil). The .sgy files were analyzed in Petrel software. The age of selected reflectors was based on the interpretation from other sources (Kowsmann et al., 1977; Contreras et al., 2010; Anjos-Zerfass et al., 2013; Morales et al., 2017).

Ten lines of sub-bottom records using a 3260 Knudsen 3.5 kHz profiler (Fig.1) were obtained during a cruise onboard R.V. Alpha Crucis, between 30 November and 05 December 2017; Knudsen-proprietary .keb files were transformed into the .sgy format and analyzed using Meridata SVIEW and MDPS softwares. Additionally, the same survey lines were performed with a Teledyne RDI 75 kHz ADCP for water current. Magnitude and direction of current speeds were filtered for the removal of spikes, gridded using the natural neighbor interpolation, and vector maps of currents were performed for surface and 200 meters water depth, using software Surfer® Version 13.

Box-core samples were taken on the previously cited cruise as well as in another cruise, held in 2007, onboard R.V. Prof. W. Besnard (University of São Paulo, Brazil). The location of the samples is presented in Table 1 and Fig.1. Each box-core was sub-sampled continuously at intervals of 2-cm and each sub-sample was kept frozen for later freeze-drying. Only the surface (0-2 cm) sections were analyzed for this study.

The grain-size analysis was performed in sediment samples, after the removal of the calcium carbonate fraction with a solution of 10% HCl, removal of organic matter with a solution of 30% of hydrogen peroxide, and the addition of a solution



of 25% sodium hexametaphosphate. Each sample was then analyzed under a Malvern Mastersizer 2000 Laser Analyser and grain-size parameters were calculated using the Gradistat ® MS-Excel macro (Blott and Pye, 2001). The sand fraction of each sample was analyzed under a stereomicroscope, in order to evaluate their siliciclastic or phosphoritic character.

Based on the optical analysis, the sandy fraction was classified according to four groups: 1) Siliciclastic with less than 20% of phosphorite, 2) Siliciclastic with 20% to 50% of oxidized phosphorite, 3) 50% - 80% of oxidized phosphorite and, 4) Oxidized phosphoritic sand.

Metals (Al, Ba, Ca, Cr, Cu, K, Fe, Mg, Mn, Mo, Sc, Sr, Ti, Zn) were analyzed in a Varian 710 ICP-OES, using the protocols of complete digestion established by the US Environmental Protection Agency 3052 Method. A detailed description of the analytical procedures is described in dos Santos et al. (2018).

Statistical analyses were performed using the software Past, version 3.20 (Hammer et al., 2001). Due to the lack of normality of the data, the correlation analyses were performed using the Spearman $\rho$ values.

For metals, we also performed a Principal Component Analysis, using standardized values ($\bar{x}= 0$, $S= \pm1$), in order to identify the elements responsible for the highest variability in the area.

Finally, we also used some Metal/Metal ratios in order to recognize the relative influence of terrigenous materials (Fe/Ca) (Arz et al., 1999; Jennerjahn et al., 2004), the origin of the terrigenous fraction (Fe/K, Mg/Al) (Singh, 2011; Govin et al., 2012; Razik et al., 2015), and the characteristics of the carbonate fraction (Sr/Ca, Mg/Ca) (Bayon et al., 2007).

## 4 Results

### 4.1 Morphology

The morphology of the RGT is widely diverse. To the south (figures 3A and 3B) the terrace is almost absent, giving place to a steep scarp (> 5º) at ca. 500 meters water depth.

In its central part (figures 3C, 3D, and 3E), the RGT is well developed, showing two smaller terraces, separated by a scarp at 300 meters (3D, 3E). The previously cited 500 meter-scarp gives place to a very rough surface, which extends down to the 1000-meters isobath. In profiles 3D and 3E, the base of the RGT is marked by the presence of a very irregular, mounded bottom, with declivities reaching up to 10o and low penetration of the acoustic signal.

To the north, the RGT gives place to a surface marked by low-amplitude oscillations of declivity (figures 3F, 3G, and 3H), marking the presence of kilometer-scale terraces, scarps, and depressions. A more detailed observation of the subbottom structures in this sector (Figure 4) shows an irregular pattern of scours cutting offshoreward inclined strata.

### 4.2 Seismic

A cross-RGT deep seismic line is shown in Figure 5. As references, we marked both the Serravallian (Mid-Miocene) Unconformity as well as the Pre-Aptian basement. At its inner part, the seismic section presents about 5 sec (TWT) of strongly fractured prograding seismic units. The basal Unit (Late Cretaceous) is marked by strong deformation and faulting.



Paleogene and Oligocene units present low amplitude and progradational terminations. To offshore, out of the RGT, these units are truncated by a strong erosional surface and a distinct set of wedge-like units is deposited.

The topmost Unit (see detail in Figure 5) corresponds to a clinoform, located at the edge of the RGT, deposited over eroded post-Serravalian (Plio-Pleistocene?) deposits. It is possible to recognize, inside this clinoform, at least seven high-amplitude reflectors. On the other hand, the top of this clinoform also presents truncation over originally toplap terminations.

The interpreted total isopach and Pre-Aptian depth map (Figures 6A and B) clearly indicate that the RGT lays on a basement high, which is projected from the continent towards ENE. In its thinner part, the RGT presents a thickness equivalent to no more than 2.5 sec (TWT), corresponding to half of the thickness of adjacent areas.

### 4.3 Hydrodynamics, Sedimentology and Geochemistry

Current speed and directions, obtained during the December 2017 cruise indicate the prevalence of a high-speed southward flow, at least from the sea surface until the 200-m isobath (Figures 7A and 7B).

A summary of the sedimentological and geochemical results is presented in Supplementary Material 1.

Grain size data reveal a whole range from sands to silts, with clay becoming a subsidiary class (Figure 8). Sandy sediments occupy the widest, less steep areas of the RGT; on the other hand, sandy silts and silts are more abundant to the north, as well as on the shelf and in water depths higher than 500 meters.

Concerning the sand characteristics, the samples from the center of the RGT are mostly rich in oxidized phosphorite (Supplementary Material 2). The amount of phosphoritic sands decreases towards the extremes of the area, and a single sample shows black (non-oxidized phosphorite) in its sandy fraction.

The analysis of the grain size distribution curves (Figure 9) indicates a wide variability of distributions, from unimodal to trimodal, poorly sorted sediments. Grain size main medians vary from 3.8 ϕ (very fine sand) to 8.8ϕ (very fine silt). There is a general trend of fining grain size towards deeper areas. Samples located in the south, between 300 and 500 meters, present a conspicuous mode at fine to very fine sand.

Table 2 presents the Spearman ρ correlation values between elements, percentage of sand and clay, and water-depth, in which it is possible to recognize four clusters, i.e., Ca and Sr, Al and Sc, Ba and Cu, and Cr, Cu, Fe, K, Mg, Mn, Ni, Ti, and Zn.

The first two components of the Principal Component Analysis account for 61.5% of the total variance (Figure 10). The first component (42.1% of the total variance) is highly influenced by the contents in Fe, Ni, and Cr, and, secondarily, by the contents in K, Ti, and Zn. The second component (19.4% of the total variance) opposes Ca and Sr against Ba and Cu.

Most of the samples show Fe/Ca ratio lower than 0.5 (Figure 11), indicating the prevalence of biogenic sediments in the area. The highest Fe/Ca ratio values are present on the shallower samples, as well as those located on the top of the RGT, these last associated with the presence of Fe-rich phosphorites.

The scatter plot of Fe/K versus Mg/Al is also an indication of high compositional variability. Most of the shallower samples exhibit low values of both Mg/Al and Fe/K, the lowest values corresponding to the samples with higher terrigenous input.





The scatter plot of Sr/Ca versus Mg/Ca (Figure 13) presents a V-shaped distribution of the samples, with their concentration towards Mg-enriched (located on the center of the RGT), but distinct samples displaced either towards Sr-enriched (deepest samples in the south) or detrital carbonate (shallower samples).

## 5 Discussion

### 5.1 Morphostructure and dynamics

The complexity shown by the bathymetry of the RGT reflects a series of depositional, tectonic and recent hydrodynamic events that took place in the SW Atlantic margin that started during the rift phase and persists until now.

The RGT is sustained by a basement high, probably associated with the extension of the Torres Arch, over which a sequence of sediments, from the Aptian to the Plio-Pleistocene, was deposited. The presence of this high makes the RGT a unique feature on the SW Atlantic upper slope terraces since none of the others seems to be sustained by basement projections (Viana and Faugères, 1998; Kowsmann and Carvalho, 2001).

The topmost strata present a clinoform, which upper surface is truncated and presents a sequence of scours with amplitudes of meters (Figure 5). Differently than proposed by Zembruscki (1979), these scours present a pattern which is parallel to the isobaths and cannot be associated with paleo-drainages. We propose that this truncation is associated with the high speed of the Brazil Current; current velocities have never been measured before our work, but the competence of the Brazil Current has been already attested previously to the north of the area (Viana et al., 2002; Duarte and Viana, 2007; Biló et al., 2014; Schattner et al., 2016).

### 5.2 Sediment source and distribution

Both grain-size and sediment composition data indicate that different sources contribute with sediments to the RGT. In the grain-size, this is characterized by the presence of distinct modes and a wide grain size variability, represented by the large spectrum of medians; concerning chemical composition, wide variability in major and minor elements is also evident.

The presence of phosphorite grains also indicates that the reworking of authigenic sediments may also be an important source of the sediments, especially between the 300 and 500-meters isobath.

Concerning the contribution of a biogenic fraction, the prevalence of values of Fe/Ca lower than 0.5 indicates that biogenic constituents are also an important source of the sediments in the area. These values are lower than those reported for the dry conditions of NE Brazil (Jennerjahn et al., 2004), indicating that the direct input of modern sediments from the adjacent continent is limited and probably restricted to the shallower samples. Nevertheless, our values are compatible with those reported by Govin et al. (2012), to the same area of the SW Atlantic.

Still, concerning the biogenic fraction, it is noticeable that, despite the prevalence of Mg-enriched carbonates, some samples exhibit a higher content of Sr, suggesting the partial occurrence of aragonite and detrital carbonates (Figure 13). The



prevalence of Mg-enriched carbonates suggests the occurrence of gas escape in the RGT, aspect that was already reported to the south (Miller et al., 2015) and to the north (dos Santos et al., 2018) of the study area.

A scatter plot of the Fe/K ratio versus the grain size median (Figure 14) shows a wide range in both parameters. The samples with higher Fe/K ratios and coarser grain size correspond to the phosphorite-rich sediments. A distinct linear trend is associated with finer medians, indicating different degrees of weathering in the sediment source areas (Razik et al., 2015). Worth to note that we cannot compare our results directly with the paper by Razik et al. (2015) since those authors used the whole grain-size distribution mean. In the RGT, the presence of sediments with very high Fe/K values might be associated with post-depositional processes, related to the formation of ferruginous phosphorites.

Concerning the Principal Component Analysis, the dispersal of samples towards different trends reveals the complexity of sources of sediments in the area. Apart from the already cited biogenic fraction-enriched samples (higher Ca and Sr), other sources are recognized. Relative enrichment of Cu and Ba can be associated with higher productivity (Von Breymann et al., 1990; Kumar et al., 1996) which is promoted by a shelf-break upwelling that is frequent in the area (Brandini, 1988, 1990). Finally, the Fe, Ni, and Cr-enriched samples are associated with the presence of Fe-enriched phosphorites (Compton and Bergh, 2016; González et al., 2016).

Considering the aspects of the geological inheritance of a feature sustained by a basement high, the mixture of both lithogenic, biogenic, and authigenic sediments, and the reshaping of the morphology, given by the action of the Brazil Current on its shallower part, we recognize that the RGT consists of a unique feature, which is distinct than all of the other slope terraces of the SW Atlantic.

## 6. Conclusions

The Rio Grande Terrace (RGT) is the most conspicuous upper slope terrace of the SW Atlantic margin. Its origin is associated with the deposition of Late Cretaceous to Cenozoic sediments over an uplifted volcanic body. Its Late Cenozoic (Pleistocene?) corresponds to a clinoform that is truncated on its top. This truncation, marked by several isobaths-oriented scours, may be associated with the strong flow of the Brazil Current.

Surface sediments reveal a complex mosaic of sources and types, varying from fine sands do very fine silts, with unimodal to trimodal distributions. The non-biogenic composition is also extremely variable and includes siliciclastic sediments and authigenic phosphorites. In this sense, the complex interaction of geological inheritance, sediment sources, and hydrodynamics led to the development of the RGT, a unique feature in the SW Atlantic.

## Acknowledgments

The authors wish to thank the crew and researchers who participated in the Nov-Dec 2017 Survey onboard R.V. Alpha Crucis. This work was funded by the São Paulo Science Foundation (FAPESP, grant 2016/22194-0). FAPESP also grants



the collaboration between MM de M and US (grant 2017/50191-8). MM de M acknowledges the Brazilian Research Council (CNPq), for the Research Grant 303132/2014-0. CNPq also supports the collaboration between MM de M and FJL (grant 401041/2014-0).

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

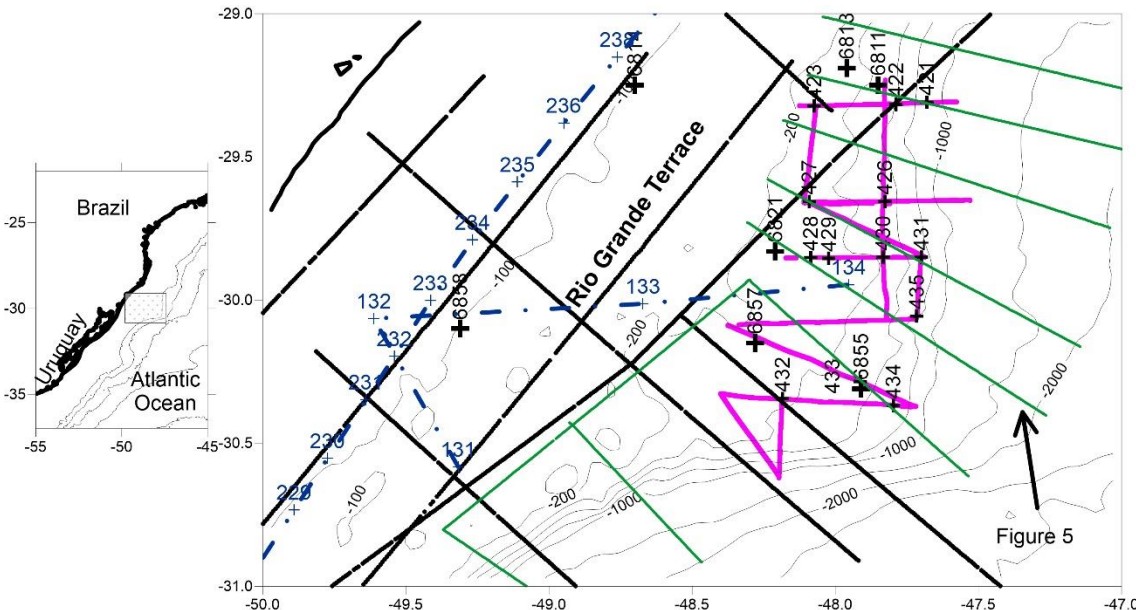

**Figure 1: Location of the study area, seismic refraction lines (Kowsmann et al., 1977, dashed blue), deep seismic reflection lines**
10  **(black), echosounding lines (green), chirp and ADCP lines (magenta), and samples analyzed in this work.**



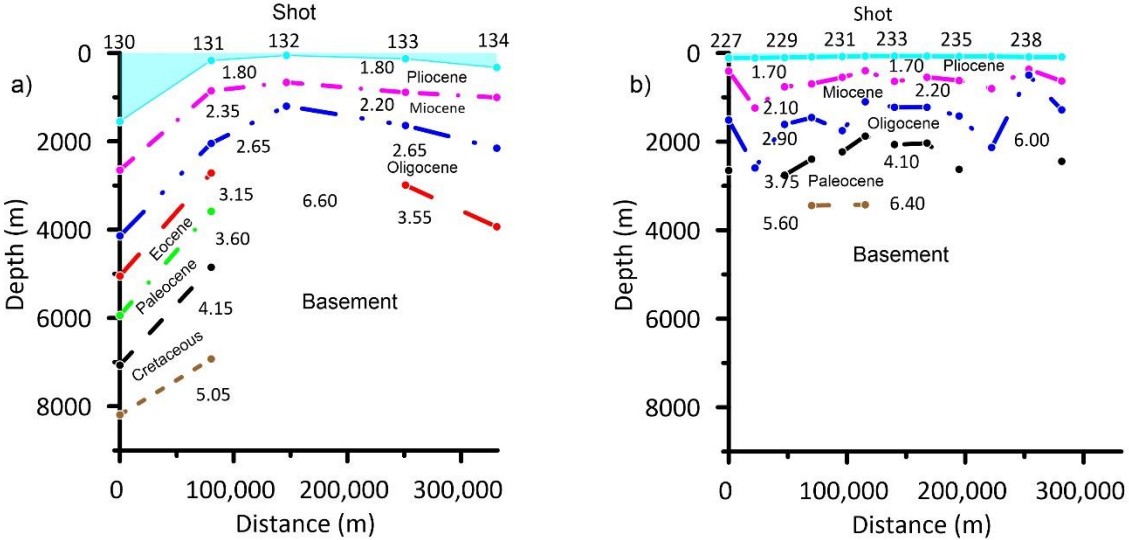

**Figure 2: Interpretation of seismic refraction lines as presented in Kowsmann et al. (1977). For position of the shooting lines refer to Figure 1.**





**Figure 3: Bathymetrical (left) and declivity profiles of the echosounding lines provided by the Brazilian Navy. Location of each profile is shown in the topmost figure.**





**Figure 3: (cont.)**



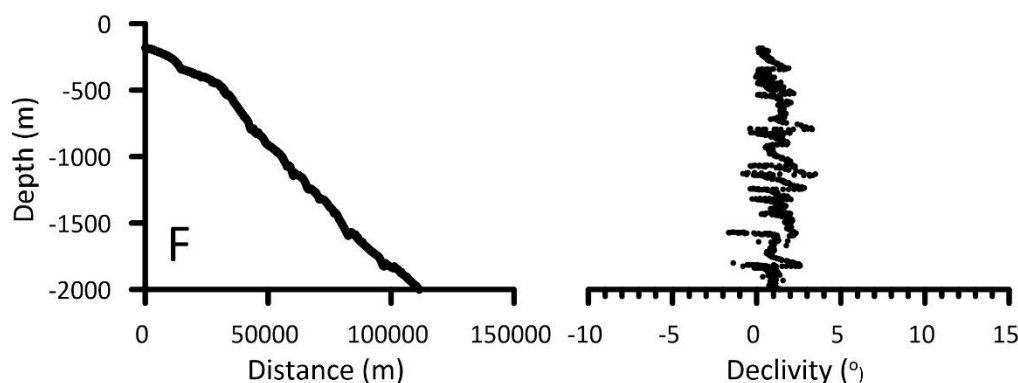

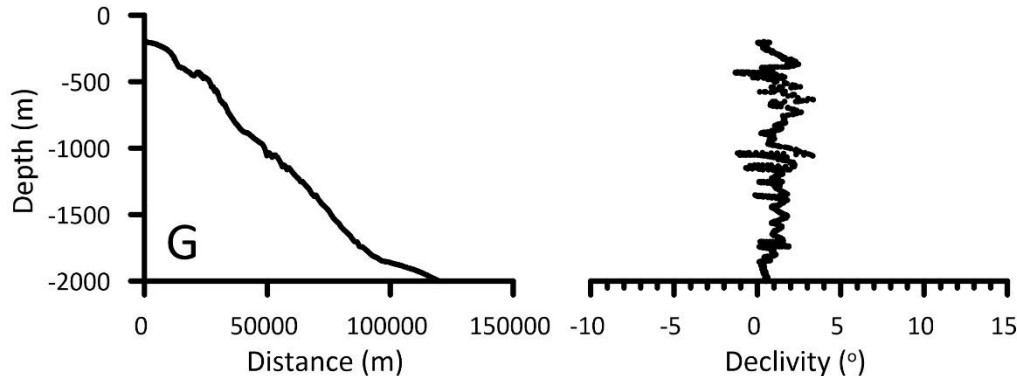

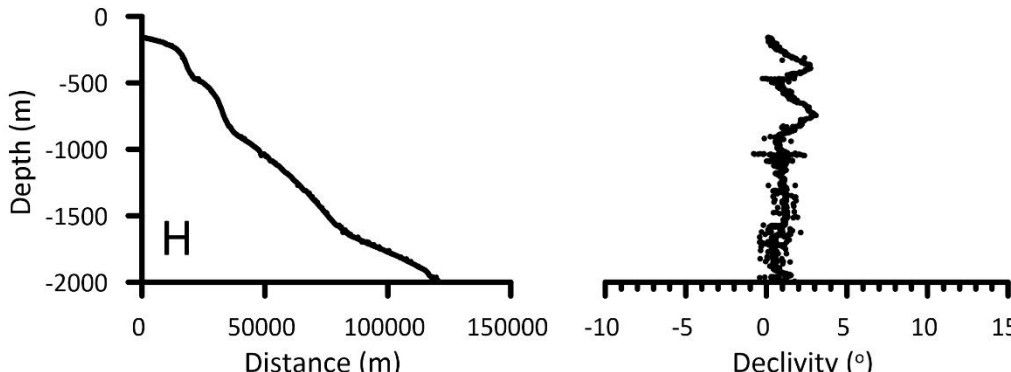

**Figure 3: (cont.)**



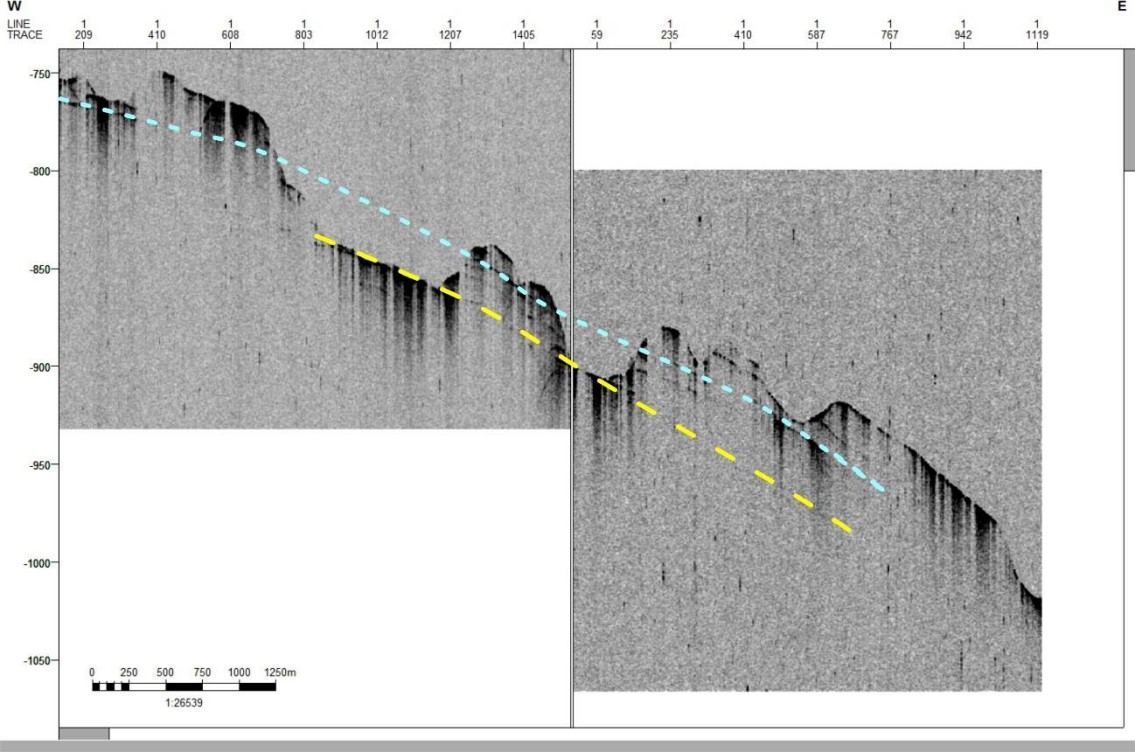

**Figure 4: Chirp line from the northern sector of the RGT, showing the truncation of previously deposited prograding sediments. Light blue and yellow dashed lines are shown as references of the continuation of the strata.**



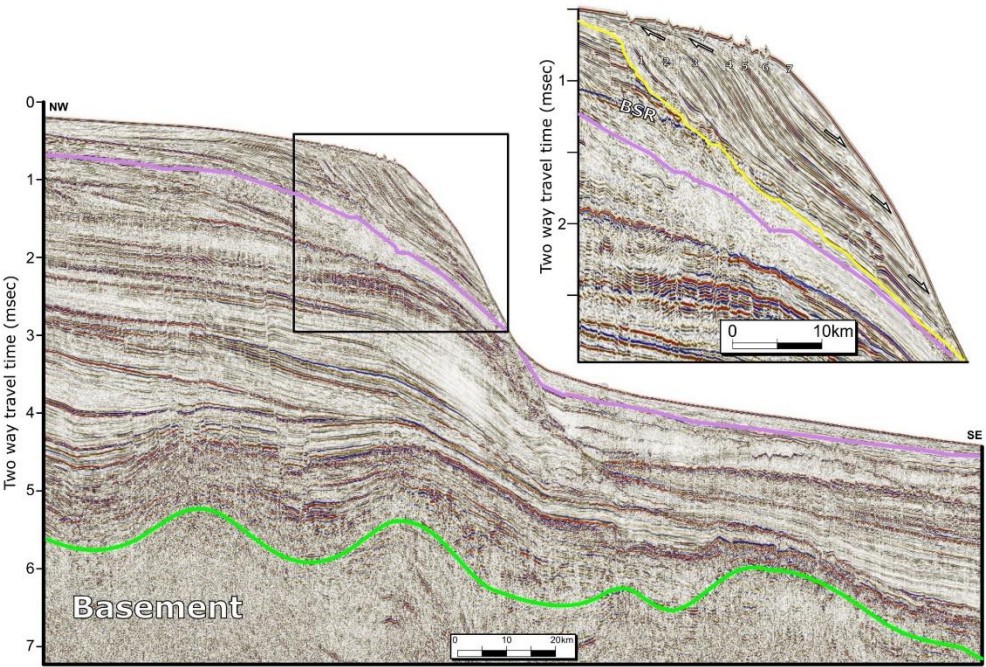

**Figure 5: Deep seismic reflection line showing the whole post-Aptian sequence. Light green and magenta lines indicate the basement and Serravalian (Middle Miocene) discontinuities. The zoomed part shows a clinoform, truncated in its uppermost part. Internal high-amplitude reflectors probably reflect the limit of Pleistocene climatic cycles.**




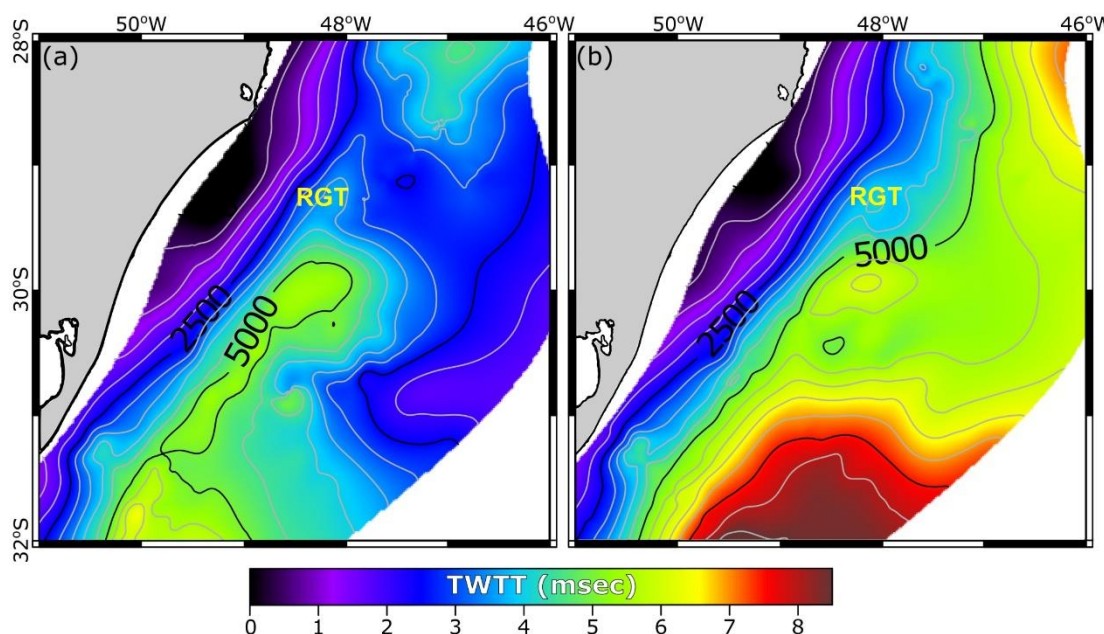

**Figure 6: Maps of sediment isopach (left) and pre-Aptian depth (values in msec)**

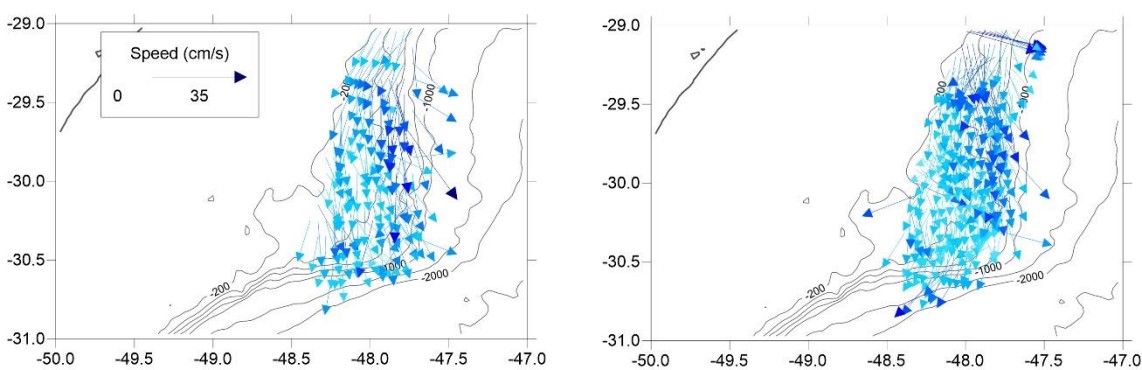

**Figure 7: Vector maps of surface (left) and 200-meters (right), current speed and direction, interpolated from ADCP data collected in a survey held in November-December 2017.**



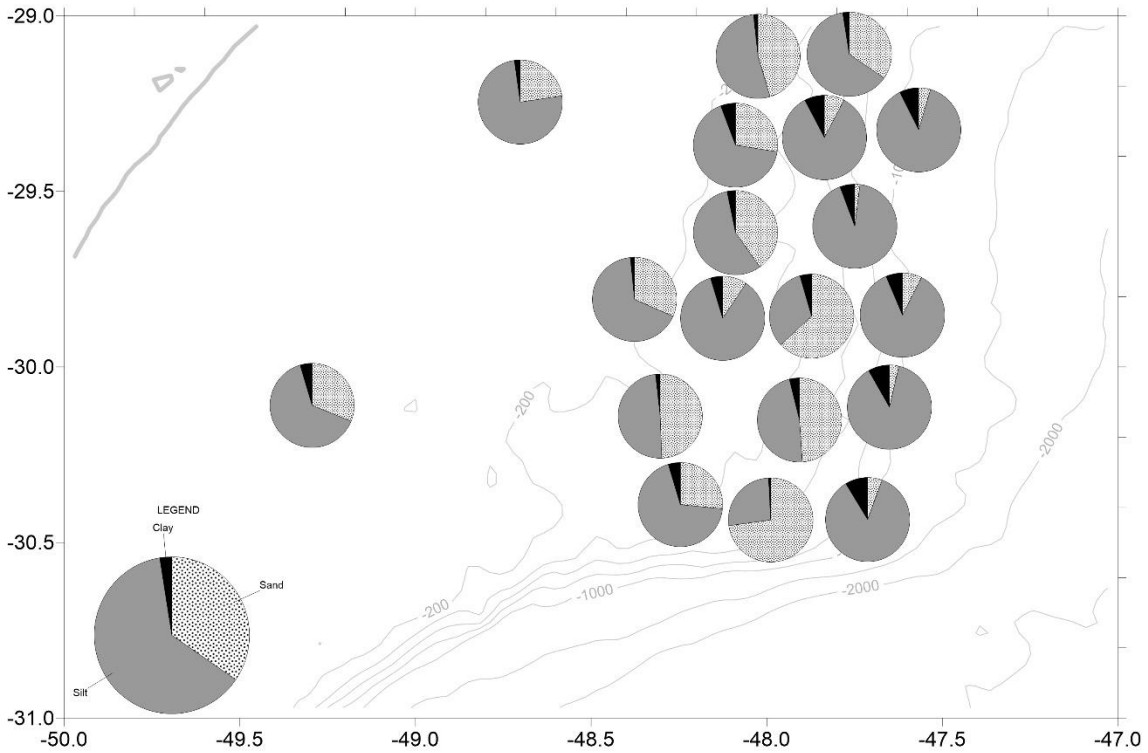

**Figure 8: Distribution of pie-charts of sand-silt-clay in the study area.**



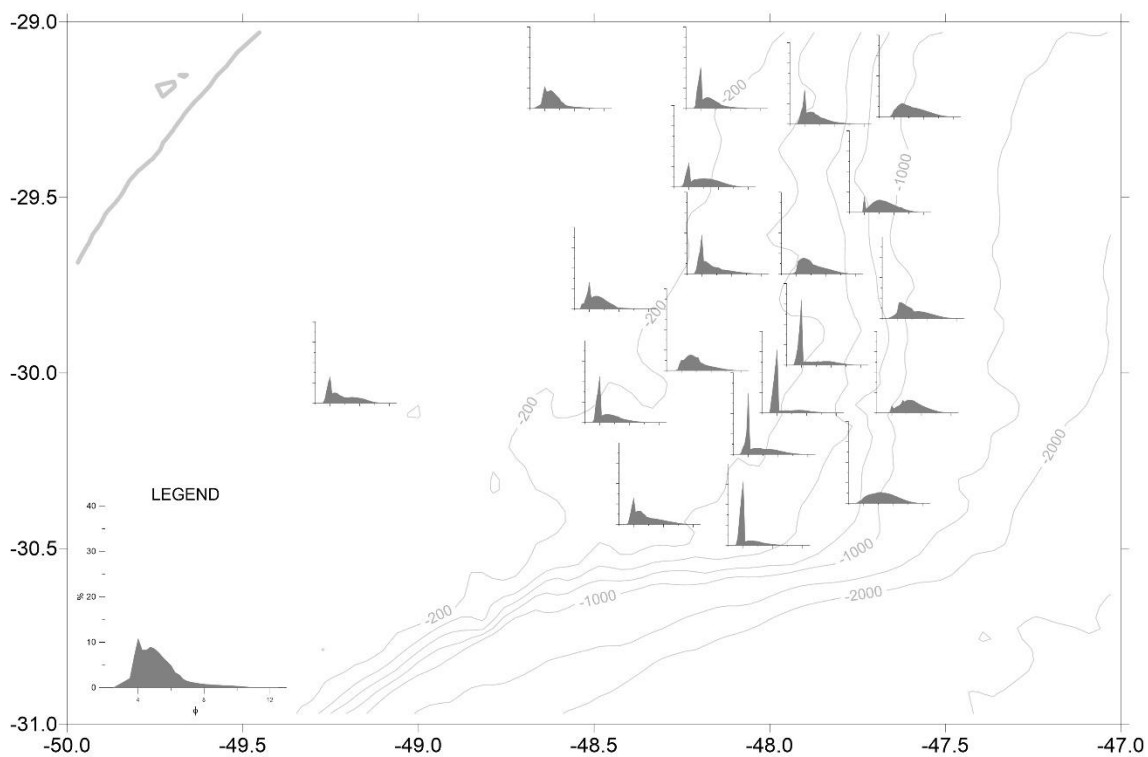

**Figure 9: Grain-size distribution curves of the samples collected in the study area.**

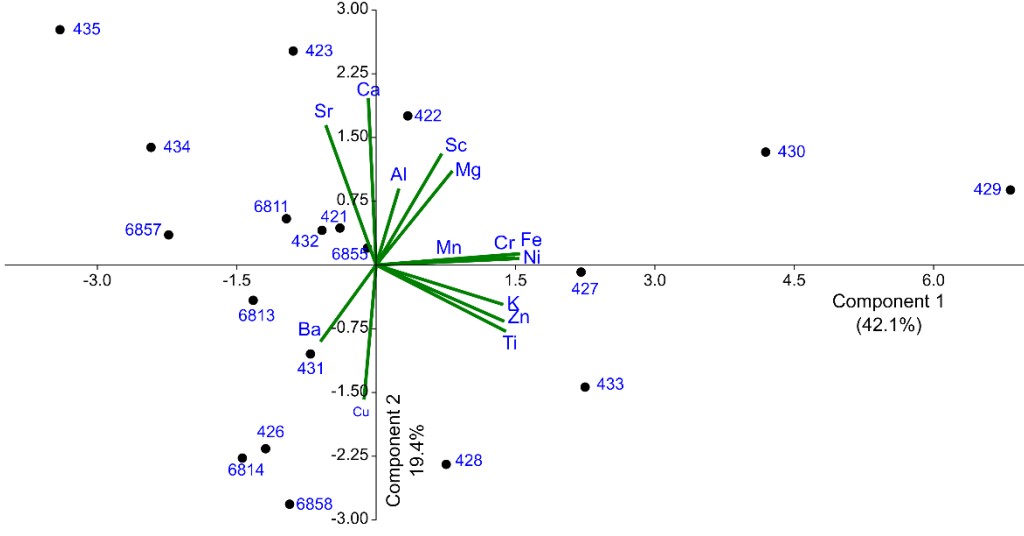

**Figure 10: Scatter plot (samples and variables) of the two first components of the Principal Component Analysis.**





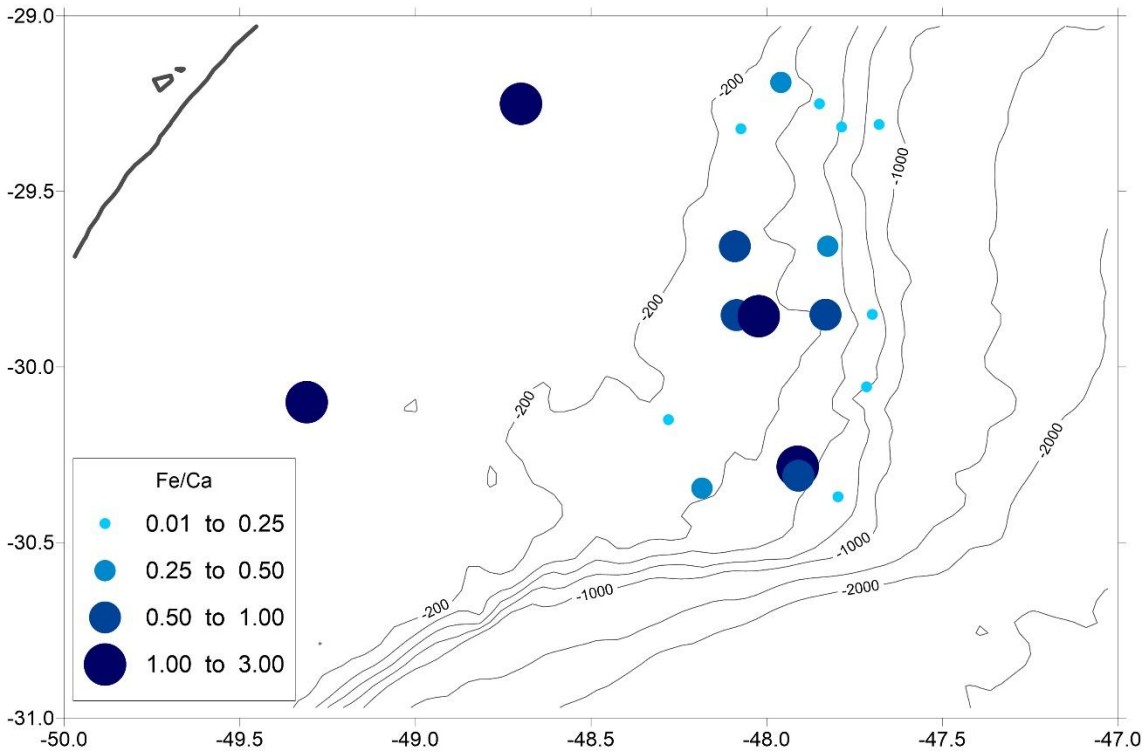

**Figure 11: Distribution of the Fe/Ca ratio in surface sediment samples of the study area.**



**Figure 12. Scatter plot of Fe/K versus Mg/Al ratios of the surface sediment samples in the study area**





**Figure 13. Scatter plot of the Sr/Ca versus Mg/Ca ratios of the surface sediments in the study area. Position of the end-members is based in Bayon et al. (2007).**




**Figure 14. Scatter plot of Fe/K ratio versus grain size median (in φ). The blue line encompasses the phosphorite-rich sediments.**

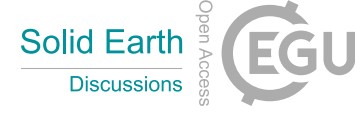

**Table 1. Location of the sediment samples used in this work**

| Station | Latitude | Longitude | Depth (m) |
|--------:|---------:|----------:|----------:|
| 421 | -29.31 | -47.68 | 888 |
| 422 | -29.32 | -47.79 | 688 |
| 423 | -29.32 | -48.07 | 234 |
| 426 | -29.66 | -47.83 | 534 |
| 427 | -29.66 | -48.09 | 344 |
| 428 | -29.85 | -48.09 | 358 |
| 429 | -29.86 | -48.02 | 348 |
| 430 | -29.85 | -47.83 | 483 |
| 431 | -29.85 | -47.70 | 800 |
| 432 | -30.34 | -48.18 | 332 |
| 433 | -30.28 | -47.91 | 490 |
| 434 | -30.37 | -47.80 | 746 |
| 435 | -30.06 | -47.72 | 797 |
| 6811 | -29.25 | -47.85 | 506 |
| 6813 | -29.19 | -47.96 | 299 |
| 6814 | -29.25 | -48.70 | 103 |
| 6821 | -29.83 | -48.21 | 232 |
| 6855 | -30.31 | -47.91 | 500 |
| 6857 | -30.15 | -48.28 | 240 |
| 6858 | -30.10 | -49.31 | 100 |

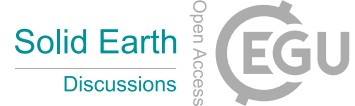



**Table 2. Spearman ρ correlation of the grain size and metals analyzed in this work**

| | Depth | Sand | Silt | Clay | Al | Ba | Ca | Cr | Cu | Fe | K | Mg | Mn | Ni | Sc | Sr | Ti | Zn |
|---|---|---|---|---|---|---|---|---|---|---|---|---|---|---|---|---|---|---|
| Depth | | 0.031 | 0.043 | 0.004 | | | 0.015 | | | | | | | | | 0.040 | | |
| Sand | -0.483 | | 0.000 | 0.000 | | | | 0.011 | 0.005 | | 0.017 | | | | | | | |
| Silt | 0.457 | -0.986 | | 0.000 | | | | 0.016 | 0.003 | | 0.024 | | | | | | | |
| Clay | 0.609 | -0.827 | 0.762 | | | | 0.003 | | 0.031 | | | | | | | 0.012 | | |
| Al | | | -0.182 | | | | | | | | | | | | 0.000 | | | |
| Ba | | | | | | | | | 0.033 | | | | | | | | | |
| Ca | 0.549 | | | 0.644 | | | | | | | | | 0.042 | 0.000 | | 0.000 | | |
| Cr | | 0.568 | -0.544 | | | | | | 0.000 | 0.000 | | | | 0.000 | | | 0.001 | 0.002 |
| Cu | | -0.621 | 0.647 | 0.495 | | 0.491 | | | | | | | | | 0.002 | | | |
| Fe | | | | | | | | 0.912 | | | 0.000 | 0.035 | 0.014 | 0.000 | | | 0.000 | 0.000 |
| K | | 0.539 | -0.516 | | | | | 0.800 | | 0.882 | | 0.009 | | 0.000 | | | 0.000 | 0.009 |
| Mg | | | | | | | | | | 0.486 | 0.582 | | 0.003 | | | | | |
| Mn | | | | | | | 0.470 | | | 0.553 | | 0.642 | | 0.007 | | | 0.030 | |
| Ni | | | | | | | | 0.860 | | 0.891 | 0.746 | | 0.598 | | | | 0.000 | 0.000 |
| Sc | | | | | 0.837 | | | | -0.658 | | | | | | | | | |
| Sr | 0.475 | | | 0.561 | | | 0.946 | | | | | | | | | | | |
| Ti | | | | | | | | 0.696 | | 0.767 | 0.742 | | 0.498 | 0.798 | | | | 0.000 |
| Zn | | | | | | | | 0.667 | | 0.730 | 0.584 | | | 0.856 | | | 0.793 | |