# Peer review of "The building, shaping, and filling of an Upper Slope Terrace: the Rio Grande Terrace, SW Atlantic"

_Solid Earth, 2018_

## Referee Comment (RC1) · Anonymous Referee #1 · 4 Apr 2019

Dear Editor, I send the review of the manuscript entitled "The building, shaping, and filling of an Upper Slope Terrace: the Rio Grande Terrace, SW Atlantic"by Michel Michaelovitch de Mahiques et al. #SE2018_140

This study combines deep to shallow seismic profiles, echo-sounding and surface sediments in order to analyze the origin, evolution, and recent sedimentation of the Rio Grande Terrace, an upper slope terrace of the SW Atlantic. The study provides a nice case study of an area relevant for oceanographic processes and development of erosive features produced by bottom currents. The authors face a very interesting topic however some main issues have to be solved before the manuscript can be accepted

for publication on SE. Overall, the manuscript is well organized, but the Result and Discussion sections need to be improved significantly, in order to better support the interpretation and geological significance of the Rio Grande Terrace (RGT) and main conclusions. In my opinion, at present the results and discussion do not support the main conclusions ( the RGT is a unique morphological feature on the SW Atlantic upper slope terraces because of the role played by geological inheritance: occurrence of a basement high). The authors "link" the occurrence of a basement high with "recent" erosional processes affecting the RGT likely related to the activity of the Brazil Current but a complex history regarding the development of the RGT, depositional sequences related to sea level changes is suggested by seismic profile shown in figure 5 and should be considered both in the results and discussion section. Moreover, the spatial relationship between the basement high and the terrace is not clear. As well as the spatial relationship between the RGT and the continental shelf is not clear, both on the maps and the seismic profiles.

My recommendations are: 1) the English language should be checked in order to correct some minor inaccuracies and clarify some sentences; 2) In the results section the morphology and seismic stratigraphy sections have to be improved in order to better support illustration and description of main morphological and stratigraphic features,. The extent of the RGT, of the continental shelf and basement high have to be clearly indicated on the maps (Fig. 1 and 3) and seismic profiles (Fig. 4 and 5). More seismic profiles showing the internal architecture of the terrace and their differences with adjacent areas have to be added. Information regarding very high resolution seismic profile have to be improved and provided in a separated dedicated section. As well as the erosive features described in the manuscript (i.e., the scours) have to be mapped in order to show their spatial pattern.

3) In the discussion., section 5.1 (Morphostructure and dynamics) needs major improvements considering the possible role of sea level changes in development of plio-pleistocene units and erosive surfaces. I think that this aspect is mandatory for the

interpretation of the origin of the erosional terrace. Even if I am not a specialist of the Brazilian margin, I think that the authors might refer to the seismic stratigraphy shown Correa da Camara Rosa et al. (High-Frequency Sequences in the Quaternary of Pelotas Basin (coastal plain): a record of degradational stacking as a function of longer-term base-level fall) and more in general to the work of Hernández-Molina et al. / Marine Geology 378 (2016) 333–349. Sequence APD4 of the Pelotas basin show the same pattern of seismic reflections; it is a plio-pleistocene composite sequence thus the authors should consider the role of sea level changes in their interpretation and discussion. Thus I strongly suggest to consider other explanations of main findings because at present it seems that the results are over-Interpreted. Moreover, I think that Section 5.2 (sediment source and distribution) of the discussion should be rewritten in order to make a more effective discussion. At present it is not well organized, thus I suggest to the authors to explain the results describing the significance of their findings in light of known literature, focusing on separate points as: sediment types, source etc.

I made some minor comments (see the attached pdf).

Figures Fig. 1: Improve location map: add indication of the continental shelf and slope; add coastal features and names Fig. 3: add indication of the continental shelf, continental slope and RGT Fig. 5: add indication of the continental shelf and RGT; add line drawing Fig. 6, 7, 8, 9 and 11: add indication of the continental shelf, continental slope and RGT I hope that my comment and suggestions will help the manuscript. Overall, I think that the manuscript can be accepted after major improvements.

Please also note the supplement to this comment:
https://www.solid-earth-discuss.net/se-2018-140/se-2018-140-RC1-supplement.zip

---

## Author Comment (AC1) · 9 Apr 2019

After discussing with the co-authors, we understood that the amount of modification is substantial and we would need to withdraw the paper for an eventual later submission. Indeed, we would need to cut almost half of the paper and focus on a single aspect. On behalf of the co-authors, thank the Editor and Reviewer for their kindness and apologize for our decision.